# Orthogonal Set of Indicators for the Assessment of Flexible Pavement Stiffness from Deflection Monitoring: Theoretical Formalism and Numerical Study

**Jean-Michel Simonin** [1,*] **, Jean-Michel Piau** [1] **, Vinciane Le-Boursicault** [2] **and Murilo Freitas** [1]

1   MAST-LAMES, Campus de Nantes, Univ Gustave Eiffel, F-44344 Bouguenais, France;
    jean-michel.piau@univ-eiffel.fr (J.-M.P.); murilo.freitas@univ-eiffel.fr (M.F.)
2   Ministry of Armed Forces France, Base Aérienne 125, USID-Bureau MRTT, Route du Camp d'Aviation,
    F-13800 Istres, France; vinciane.le-boursicaud@intradef.gouv.fr
*   Correspondence: jean-michel.simonin@univ-eiffel.fr

**Abstract:** The monitoring of pavements along roads is generally based on the use of indicators directly derived from measurements. More specifically, the bearing capacity of pavements is often simply deduced from either the maximum deflection value measured or the difference between two measured values along the deflection basin. This paper proposes a methodology to define a set of orthogonal indicators adapted to the structure being evaluated. This methodology is presented for deflection measurements recorded on a flexible pavement simulated by the Burmister model and consists of searching the weighting functions to calculate different indicators as linear forms of the deflection bowl. Weighting functions are defined for each indicator in order to maximize its sensitivity to a given structural parameter without being sensitive to the other structural parameters. The paper presents the various steps involved in constructing the indicators. A numerical example of an application shows that variations of each indicator follow the Young's modulus variations specific to this indicator. Several extensions of this method are also introduced for other mechanical models or instrumented pavements.

**Keywords:** monitoring; indicator; deflection; pavement survey; orthogonal method

## 1. Introduction

Roads constitute the main means of communication throughout the world and are being used constantly to transport people and goods [1]. Maintaining road networks in good condition requires efforts despite the decrease in financial investments allocated to road maintenance. In particular, these efforts entail greater monitoring in order to plan for the priority maintenance work to be carried out and decide where, when, and how to intervene.

Several techniques, protocols, and devices can be used to monitor the structural pavement condition at various scales along a road section. The relatively recent project called TRIMM [2] has described the most widely used techniques including the visual or automatic detection of defects (roughness, cracking, rutting, etc.), deflection and radar measurements and coring [2,3]. These techniques provide different information, which can be complementary with other techniques to produce an accurate diagnosis of the condition of a given road. Such information can also be used like 'Russian dolls' in a nested configuration. Techniques that lend themselves to large-scale monitoring make it possible to focus on coring sites at a more local level, in yielding more precise information on the state of materials in place and the interfaces between layers.

### 1.1. Use of Deflection Measurements

Among the various monitoring methods available, deflection measurement is the worldwide standard used to evaluate pavement structures. Pavement deflection bowls

are estimated by applying a force of known or measured value to the pavement surface and then measuring the corresponding vertical displacement at the road surface. On a large scale, deflection is employed to assess both the global stiffness of pavements and the relative stiffness provided by the soil and structural layers of pavements. Based on deflection measurements, project owners are able to define homogeneous sections, likely to fall under the same maintenance strategy. On a local scale, measurements can be performed to determine the stiffness of the main pavement layers as well as the behavior of the interfaces lying between them. This information can establish the reinforcement techniques to be applied. Such a maintenance strategy is also applied to managing large airport pavements, which are often treated in homogeneous zones.

Deflection basin measurements can be processed in two main ways. The first uses what may be called 'quick-to-calculate indicators', which include the maximum deflection, Rd (radius of curvature), BLI, MLI, and LLI indicators [4–6], all well-known in road monitoring. Le-Boursicault [7] proposed a synthesis of these conventional indicators (see Table 1); these are based on simple and various combinations of the surface deflection measured at different distances from the load. They are able to provide an estimate of the stiffness of the various pavement layers. The alternative method consists of back-calculating layer stiffness by seeking the best match between the measurements and the simulation of deflections from a pavement model. Typically, for flexible pavements, the model chosen for this step is Burmister [8], as widely used in pavement design [9].

**Table 1.** Conventional indicators used to interpret deflection measurements (as synthesized from Le Boursicault [7]).

| Index | Definition | Comments | References |
|---|---|---|---|
| $D_0$: Maximum Deflection | $D_0 = D_{max}$ | Affected by all layers | [2–7,10–16] |
| $D_i$: Deflections | Deflection measurement recorded by sensor #i or at "i" millimeters from the center of the plate | | [5–7,10,11] |
| $R_oC$: Radius of Curvature | Second derivative of the deflection basin at the maximum deflection Calculation method depending on the device | Sensitive to both the base layer and interface | [4,7,12,13] |
| Rd: | $Rd = R_oC \times D_0$ | Sensitive to platform variations for flexible pavements | [4,7] |
| BLI: Base Layer Index or SCI: Surface Curvature Index | $BLI = D_0 - D_{300}$ | More sensitive to surface layers | [10,11] |
| MLI: Middle Layer Index or BDI: Base Damage Index | $MLI = D_{300} - D_{600}$ | More sensitive to base layers | [10,11] |
| LLI: Lower Layer Index or BCI: Base Damage Index | $LLI = D_{900} - D_{600}$ | More sensitive to both base and foundation layers | [10,11] |

The method based on the indicators shown in Table 1 makes it possible to generate pavement condition diagrams along a road section for purposes of comparison. However, an interpretation of indicator values in terms of pavement defects is not highly accurate. Conversely, the second method (transformation of the deflection bowl into layer's stiffness modulus) offers a more explicit interpretation of the measurements, yet often faces the difficulty of leading to several possible 'back-calculated solutions', which can differ significantly from one another. Next, an operator is required to assist with the diagnosis and produce consistent variation curves of the interpreted data along a given road section.

The topic here deals with an improvement to the first approach, with our goal being to build a set of stiffness indicators $\{I_l\}$ ($l \in \mathbb{N}$), each of which relates to either a specific pavement layer parameter—e.g., Young's modulus—or an interface state between two

layers. These new indicators offer the advantage of a direct interpretation based on a physical parameter. Moreover, these indicators can only be sensitive to a single physical parameter, which will prove to be helpful in detecting and identifying pavement defects. In contrast, the conventional indicators, as indicated in Table 1, are sensitive to several physical parameters of the pavement structure.

### 1.2. Recall of the Various Means for Conducting Deflection Measurements

A large number of devices have been developed for deflection measurement. Let us distinguish the three types of measurement principles. The most common is to apply a load at a fixed point and measure the deflection at various distances from the load. This principle is applied using the falling weight deflectometer (FWD) [10,11], which has been continuously developed and improved since the 1960's. The main difficulty with FWDs lies in obtaining sufficient measurement density along a given route. In order to solve this problem, curviameters [12,13] and deflectographs [14–16] were developed before the end of the last century. Their principle entails using a heavy twin load on the rear axle and measuring the deflection bowl at a fixed point on the road when the load is rolling. Ingenious systems have been developed to place the measurement device at a fixed point on the pavement to record a measurement and then place the device further along the route, in general with a regular spacing of 5 m. The measurements are collected almost continuously at speeds of between 3 and 18 km/h along the inspected route. While the accuracy of these devices is lower than that of FWDs, they still provide an effective overview of the structural condition of a route. Since the beginning of this century, much research has been conducted into developing a new generation of devices combining the use of a rolling load and a system for measuring deflections at fixed distances from this load [17,18]. For example, the traffic speed deflectometer (TSD) uses multiple velocimeters to estimate the deflection basin formed in front of the rear axle. Such measurements can be recorded every 10 m at traffic speeds of up to 100 km/h [19,20], thus resolving the safety and inconvenience issues encountered by other devices.

Regardless of the device, pavement monitoring yields a succession of deflection bowls $\{w_{meas}\}(s_m)$. It is assumed that these data are correlated with the vicinity of successive points $O_m$ of abscissa $s_m$ ($m \in \mathbb{N}$) along the inspected road. The interval between 2 successive points $O_m$ depends on the device used and varies from 5 to several tens of meters. On the local scale of a deflection bowl, the $w_{meas}$ values are considered to be dependent on the algebraic distance, denoted, $x$, between the point(s) of application ($P_L$) of the deflectometer load and the point(s) ($P_M$) of the deflection measurement. The $w_{meas}$ values can thus be denoted $w_{meas}(x; s_m)$ or $w_{meas}(x_i; s_m)$ with $x_i \in \mathcal{M} = \{x_j; j = 1, \ldots, n_{\mathcal{M}}\}$.

For FWD and TSD devices, up to $n_{\mathcal{M}} = 10$ independent sensors can be used to measure the deflection basin, while for deflectographs or the curviameter a single sensor records the deflection basins with some 100 points.

The next part of this paper will present the method for constructing the optimized indicators deduced from a measurement dataset $\{w_{meas}(x; s_m)\}$. Part 3 will then propose a numerical application of the method. Possible method extensions will suggested in Part 4.

## 2. Construction of Indicators to Assess the Individual Stiffness of Pavement Layers

The following will consider a given subset of deflection measurements $\{w_{meas}\}(s_m)$ or more simply $\{w_{meas}\}$, as defined above at the scale of a deflection bowl.

### 2.1. Pavement Model for the Determination of Indicators

The determination of indicators $\{I_l\}$ is based on the use of a mechanical model $(\mathfrak{M})$ of the pavement under study that makes it possible to simulate the measurement process from the deflection device under consideration. For the case of flexible pavements envisaged in this article, $(\mathfrak{M})$ can be chosen as Burmister's elastic multilayer model (Figure 1), which is known in general to offer an accurate description of the mechanical behavior of these

pavements The developments presented below have therefore been carried out for this particular case, but potential variations will be discussed in Section 4.

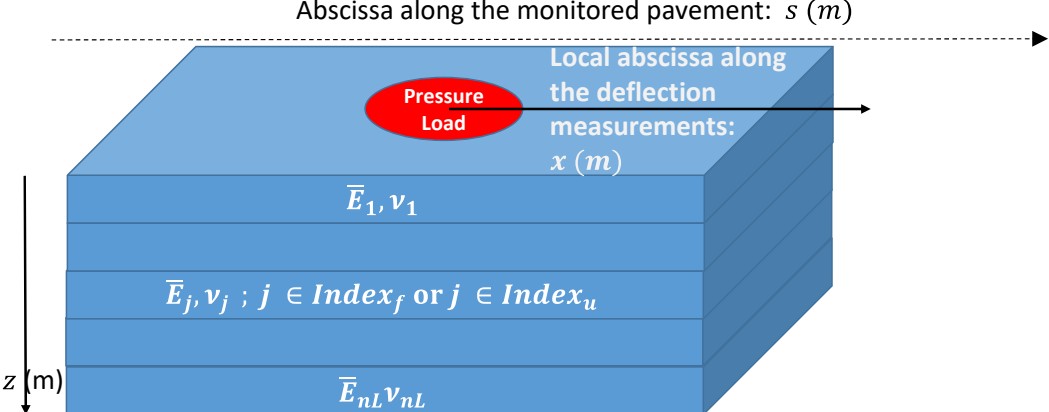

**Figure 1.** Typical elastic multi-layer structure used as a direct model for calculating the indicators. The load simulates that used for deflection measurements. The number of layers $n_L$ can differ from that of the pavements. Some model layers have a predetermined and fixed stiffness ($j \in Index_f$), while others ($j \in Index_u$) are considered with unknown values of Young's modulus, which must be specified from the deflection measurements ($n_f + n_u = n_L$).

Let us denote $n_L$ the number of model layers, which may differ from the number of actual layers constituting the pavement. Indeed, adjacent pavement layers made of similar materials may be merged into one in the model; conversely, additional thin layer(s) may be introduced to account for interface(s), with an intermediate behavior between perfectly adhesive and sliding conditions.

Let $E_j$ be the elastic stiffness modulus of layers $j = 1, \ldots, n_L$. Two sets of complementary layers can optionally be distinguished with the possible benefit of reducing the number of unknowns, consisting of those for which the $E_j$'values are considered to be given, with fixed values $\overline{E}_j$; and those for which the $E_j$'values are considered as unknowns to be determined. However, even for these latter values, we will consider them to vary in the vicinity of a given set of values, as denoted $\overline{E}_j$. Let us therefore assume for the deflection bowl at point $O_m$

$$E_j(s_m) = \overline{E}_j + \delta E_j(s_m) \tag{1}$$

where the $\delta E_j(s_m)$ values are assumed to be small compared to $\overline{E}_j$.

Let us also denote $Index_f = \left\{ j_{f,1}, \ldots, j_{f,n_f} \right\}$, $Index_u = \{ j_{u,1}, \ldots, j_{u,n_u} \}$ the corresponding sets of $j$ indices, with $n_f + n_u = n_L$ ($f$ = "fixed", $u$ = "unknown").

The other parameters of $(\mathfrak{M})$ are: the geometry and magnitude of the deflectometer load, layer thicknesses $h_j$, and Poisson's ratios $\nu_j$. All these data, including $\overline{E}_j$ are assumed to be known and highly representative of both the 'average behavior' of the pavement structure under investigation and the deflection measurement process.

Next, let us suppose for example that the set of $\overline{E}_j$ values stem from a back-calculation process carried out at some specific abscissa, $\overline{s}$, under the watchful eye of a pavement specialist, using in-depth knowledge of the pavement structure at this particular location (e.g., deflection measurements + coring data). Obviously, the choice of $\overline{E}_j$ is not unique, our approach does not rely this assumption; the most important point is to possess a good estimation of $\overline{E}_j$ values so that the differences $\delta E_j(s_m)$ between the 'true' values $E_j(s_m)$ and $\overline{E}_j$ remain relatively small.

From this context, the objective assigned to the indicators $I_l$ is to specify the 'real' stiffness moduli values $E_l(s_m)$ of the pavement layers #$l$ for $l \in Index_u$, using the measurements data and the model $(\mathfrak{M})$. This one will allow us in the following to calculate when needed the theoretical response of the pavement for any set of parameters $E_l$.

### 2.2. Proposed Indicators and Constraints for Their Determination

Let us note $w_{\mathfrak{M}}(P_M, P_L; \{E_k\}_u)$ the 'theoretical deflection function' issued from the model $(\mathfrak{M})$ for the set of Young's modulus values $\{E_k\}_u$. $(P_M)$ is the point where the (referential) load is positioned; $(P_L)$ is any point of observation at the surface of the pavement. In the case of Burmister's model, whereby deflection only depends on the distance between $P_L$ and $P_M$, this function is used subsequently in the following form

$$w_{\mathfrak{M}}(x; \{E_k\}_u) \; or \; more \; simply \; w_M(x), with \; x = \overline{P_M P_L} \tag{2}$$

From the model $(\mathfrak{M})$, let us now consider indicators $I_l$, defined as linear forms of the measured deflection, i.e.,

$$I_l(s_m) = \int_{\mathcal{M}} p_l(x) \, w_{meas}(x \, ; s_m) \, dx = p_l \cdot w_{meas} \tag{3}$$

where:

- $l \in Index_u$
- $p_l(x)$ = "Weighting functions" (or distributions) defined on $\mathcal{M}$
- $\int_{\mathcal{M}} f(x) \, dx$ = linear form for functions $f$ from $\mathcal{M}$ to $\mathbb{R}$, defined as either:
  - $\int_{\mathcal{M}} f(x) \, dx = \int 1_{\mathcal{M}}(x) \, f(x) dx,$

    $$\approx \sum_{x_i, \, x_{i+1} \in \mathcal{M}} \frac{(f(x_i) + f(x_{i+1}))}{2}(x_{i+1} - x_i)$$

    in the case of (quasi) continuous measurements, with $1_{\mathcal{M}}(x)$ being the characteristic function of the interval $[x_1, \, x_{i_{\mathcal{M}}}]$
  - Or: $\int_{\mathcal{M}} f(x) \, dx = \sum_{i \in \mathcal{M}} f(x_i)$ in the case of discrete measurements
- $f \cdot g = \int_{\mathcal{M}} f(x) \, g(x) dx$ $(= \sum_{i \in \mathcal{M}} f(x_i) g(x_i)$ in the discrete case) = scalar product of functions $f, g$ defined on $\mathcal{M}$ and related to the norm assumed to be finite: $\|f\| = \sqrt{f.f} = \sqrt{\int_{\mathcal{M}} f^2(x) \, dx}$ $(= (\sum_{i \in \mathcal{M}} f(x_i)^2)^{\frac{1}{2}}$ in the discrete case)

The linear forms depend on the chosen functions $p_l$. They are determined so as to obtain the following indicator properties:

- Indicator maximizes the sensitivity of the deflection measurements to the stiffness of layer #$l$ (condition #1).
- Indicator is "weakly" sensitive to the stiffness of the other layers #$j$ for $j \in \backslash$ (condition #2). The best case would be for indicators to be independent of the stiffness of the other layers #$j$ ($j \neq l$) (orthogonal indicator).
- The functions are imposed to have a finite norm $=< +\infty$, in avoiding infinite values for (condition #3).
- The values that is the magnitude of functions are chosen to give a direct physical meaning to the indicators (condition #4).

The construction of the functions is then derived from these conditions as well as from the approximation of the true deflections measured onsite by the function $(x)$ (Equation (2)).

### 2.3. Determination of the Weighting Functions

The abscissa $s_m$, which has no specific role here, will be temporarily omitted from the notations below. On the other hand, should it prove to be necessary, the Young's modulus values $E_k$ for $k \in Index_u$ used to calculate the deflection can be specified in $w_{\mathfrak{M}}$, in which case the function $w_{\mathfrak{M}}$ would be denoted $w_{\mathfrak{M}}(x; \{E_k\}_u)$ or $w_{\mathfrak{M}}(\{E_k\}_u)$. As a reminder, the set of values $\{E_k\}_f$ for $k \in Index_f$ is considered to be fixed, and implicitly equal

to $\{\overline{E}_k\}_f$. In considering the model $(\mathfrak{M})$ as a fairly accurate description of the true behavior of the monitored pavement when $\{E_k\}_u$ is relatively close to $\{\overline{E}_k\}_{u'}$, $(\mathfrak{M})$ can then be used to calculate the actual values expected for indicators $I_l$ and evaluate their sensitivity with respect to the stiffness of the pavement layers or interfaces.

Let us now apply $I_l$ to the theoretical deflection $w_{\mathfrak{M}}$, i.e.,

$$I_l\left(\{E_k\}_u\right) = \int_{\mathcal{M}} p_l(x)\, w_{\mathfrak{M}}(x;\,\{E_k\}_u\,)\, dx = p_l \cdot w_{\mathfrak{M}}(\,\{E_k\}_u) \tag{4}$$

In the vicinity of $\{\overline{E}_k\}_{u'}$, the sensitivity of $I_l$ to Young's modulus $E_m$, for $m \in Index_u$ can be assessed from the following derivative

$$\frac{\partial I_l}{\partial E_m} = \overline{w}_{\mathfrak{M},E_m} \cdot p_l \tag{5}$$

where: $\overline{w}_{\mathfrak{M},E_m} = \frac{\partial w_{\mathfrak{M}}}{\partial \overline{E}_j}\left(x;\{\overline{E}_k\}_u\right)$.

Consequently, according to condition #2, for indicator $I_l$ to be insensitive to the other layers would require

$$\overline{w}_{\mathfrak{M},E_m} \cdot p_l = 0 \; for \; m \in Index_u \backslash \{l\} \tag{6}$$

Furthermore, condition #1, which calls for maximizing the variations of $I_l$ with respect to those of $E_l$, amounts to identifying the function $p_l$ leading to an *extremum* of $\overline{w}_{\mathfrak{M},\,E_l} \cdot p_l$ over the set of functions satisfying conditions #2 and #3. This problem can then be solved by expressing the stationarity of the Lagrangian $\mathcal{L}(p_l, \lambda_m, \mu)$, defined as

$$\mathcal{L}(p_l, \lambda_m, \mu) = \overline{w}_{\mathfrak{M},E_l} \cdot p_l + \sum_{m \in Index_u \backslash \{l\}} \lambda_m \overline{w}_{\mathfrak{M},E_m} \cdot p_l + \mu\,(p_l \cdot p_l - C_l{}^2) \tag{7}$$

where $\lambda_j, \mu$ are constants.

By differentiating $\mathcal{L}$ with respect to $p_l$, the stationarity of $\mathcal{L}$ yields the following equation

$$\left(\overline{w}_{\mathfrak{M},E_l} + \sum_{m \in Index_u \backslash \{l\}} \lambda_m \overline{w}_{\mathfrak{M},E_m} + 2\mu\, p_l\right) \cdot \delta p_l = 0 \; for \; any \; \delta p_l \tag{8}$$

which implies that the first term between brackets equals zero, or moreover that $p_l$ belongs to the space $\mathcal{W}$ spanned by the functions $\overline{w}_{\mathfrak{M},\,E_k}$. Hence

$$p_l(x) = \sum_{k \in Index_u} \alpha_{lk} \overline{w}_{\mathfrak{M},E_k}(x) \tag{9}$$

where $\alpha_{lk}$ are coefficients to be determined.

From Equation (6), these coefficients must satisfy the following equations

$$\sum_{k \in Index_u} \alpha_{lk}\left(\overline{w}_{\mathfrak{M},E_k}.\overline{w}_{\mathfrak{M},E_m}\right) = 0 \; for \; m \neq l \tag{10}$$

Equation (10) can be interpreted as $p_l$ being "orthogonal" to any change in deflection $\overline{\delta w}_{\mathfrak{M}}$, as implied by (small) variations in Young's moduli $E_m (m \in Index_u)$ except for $E_l$ itself. As a correlation, indicator $I_l$ will be insensitive to all such variations, i.e.,

$$\begin{cases} p_l.\,\overline{w}_{\mathfrak{M},E_m} = 0 \\ I_{l,m}(\overline{w}_{\mathfrak{M}}) = I_l(\overline{w}_{\mathfrak{M},E_m}) = 0 \end{cases} \; for \; m \neq l \tag{11}$$

To complete the set of Equations (10) for the determination of coefficients $\alpha_{lk}$, let us rewrite the derivative $\frac{\partial I_l}{\partial E_l} = \overline{w}_{\mathfrak{M},\,E_l} \cdot p_l$ as

$$\frac{\partial I_l}{\partial E_l} = \sum_{k \in Index_u} \alpha_{lk}\left(\overline{w}_{\mathfrak{M},E_k}.\overline{w}_{\mathfrak{M},E_l}\right) \tag{12}$$

It is then convenient to set this quantity equal to 1, as follows

$$\sum_{k \in Index_u} \alpha_{lk} \left( \overline{w}_{\mathfrak{M}, E_k} . \overline{w}_{\mathfrak{M}, E_l} \right) = 1 \tag{13}$$

Since small variations of $I_l$ can be directly interpreted as variations of $\delta E_l$ (condition #4) in the vicinity of $\overline{E}_j$, i.e.,

$$\delta I_l \approx \delta E_l \tag{14}$$

for small values of $\delta E_l$.

Coefficients $\alpha_{lk}$ are obtained as the solution to the linear system with size $n_u \times n_u$, derived from Equations (10) and (13)

$$K\, A_l = \Delta_l \tag{15}$$

where:

$K = \left[ K_{ij} \right] = \left[ \left( \overline{w}_{\mathfrak{M}, E_i} . \overline{w}_{\mathfrak{M}, E_j} \right) \right]$ for $i, j \in Index_u$ is a symmetric square matrix

$A_l = \left\{ \alpha_{lj} \right\}$ for $j \in Index_u$ is a column vector

$\Delta_l = \left\{ \delta_{lj} \right\}$ for $j \in Index_u$ is a column vector, with $\delta_{ll} = 1$, $\delta_{lj} = 0$ for $j \neq l$.

Since in practice the number $n_u$ is small, matrix $K$ can be easily computed by estimating the derivatives $\overline{w}_{\mathfrak{M},\, E_i}$ from the finite differences

$$\overline{w}_{\mathfrak{M},\, E_i}(x) \approx \frac{w_{\mathfrak{M}}\left( x; \{\overline{E}_j + \varepsilon_i\, \delta_{ij}\} \right) - w_{\mathfrak{M}}\left( x; \{\overline{E}_j - \varepsilon_i\, \delta_{ij}\} \right)}{2\varepsilon_i} \; for\; small\; \varepsilon_j\; values\; vs.\; \overline{E}_j\; values \tag{16}$$

Lastly, solving the systems in (15) for $l$ varying from $j_{u\,1}$ to $j_{u\,n_u}$ and returning to the deflection measurements, leads to the family of indicators $I_l$ defined as

$$I_l(s_m) = \int_{\mathcal{M}} \left( \sum_{j \in Index_u} \alpha_{lj} \overline{w}_{\mathfrak{M}, E_j}(x) \right) w_{meas}(x\,; s_m) dx \underbrace{=}_{\substack{Discrete \\ case}} \left( \sum_{j \in Index_u} \alpha_{lj} \overline{w}_{\mathfrak{M}, E_j} \right) . w_{meas} \tag{17}$$

### 2.4. Variations of Indicators $I_l$ along a Given Route

In practice, the purpose of deflection studies on pavements is more aimed at detecting variations in stiffness along a given route (road, runway, etc.) than to determine in absolute terms the stiffness modulus values of its component layers. Then by setting the measured deflection basin $\{w_{meas}\}(s_{m_{ref}})$ as the reference at some abscissa $s_{m_{ref}}$, a more useful family of indicators $\Delta I_l(s_m)$ can be defined by the respective differences

$$\Delta I_l(s_m) = I_l(s_m) - I_l\left( s_{m_{ref}} \right) \tag{18}$$

where $I_l\left( s_{m_{ref}} \right)$ is the value of indicator $I_l$ for the data $\{w_{meas}\}(s_{m_{ref}})$. These indicators are better suited for back-calculation tools to indicate the variations in stiffness modulus between adjacent road sections. The diagrams of $\Delta I_l$ with $s_m$ are particularly well suited for revealing the general structural condition of a road section and its degree of homogeneity, as well as detecting its weakest areas, which may require more in-depth diagnostics and possibly the need for (preventive) maintenance work.

## 3. Numerical Applications of the Method (Theoretical Examples)

The method proposed above for the construction of 'orthogonal indicators' is illustrated in this article by means of a theoretical example, wherein the pavement is assumed to correspond to Burmister's model. This type of model is standard for pavement design or reinforcement. The pavement structure is assumed to be of the flexible type, with the

properties listed in Table 2. The reference structure corresponds to a classical pavement from the French pavement catalogue; it is composed of a bituminous concrete wearing course, with two bituminous base (sub)layers (BM$_1$, BM$_2$), lying on an unbound granular material subgrade layer (UGM) and rigid bedrock 6 m deep. Presence of such bedrock is a common hypothesis in pavement design. Indeed experiments using anchored sensors during the 1960s confirmed that deflection could be assumed null at a depth of 6 m. Large variations in pavement stiffness are considered along the simulated route. Variations lie within a realistic range of stiffnesses, as observed namely by:

- Variations in the Young's modulus of the upper base layer between 3000 and 18,000 MPa.
- Variations in the Young's modulus of the subgrade layer between 20 and 200 MPa.

**Table 2.** Theoretical example for constructing orthogonal deflection indicators. Case of a flexible pavement structure with fictitious large variations in layer stiffness.

| Material Type | Thickness (m) | Reference Structure Young's Modulus (MPa) | Variations (MPa) |
| --- | --- | --- | --- |
| BBSG | 0.06 | 7000 | |
| BM1 | 0.08 | 9000 | 3000 to 18,000 |
| BM2 | 0.08 | 9000 | |
| UGM | 6 | 50 | 20 to 200 |
| Rigid bedrock | Infinite | 55,000 | |

The calculation of weighting functions for the reference values in Table 2 has been performed here for the curviameter. Detailed results are available in the Le-Boursicault thesis, which also presents results for the deflectograph device. Figure 2 shows the simulated deflection basins 'recorded' by the curviameter. Both the measured deflection and its first derivative are assumed to equal zero at 3 m in front of the load. Variations in the Young's modulus of the base layer induce small variations in the maximum deflection whereas variations in the subgrade stiffness lead to significantly greater deflection variations over the entire deflection basin.

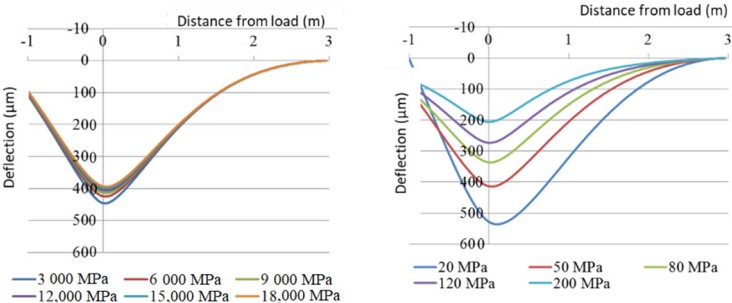

**Figure 2.** Theoretical curviameter deflection basins; **left**: stiffness variations in the bituminous base layer; **right**: stiffness variations in the subgrade layer.

The weighting functions relative to indicators $I_{BM1}$ and $I_{UGM}$ (Figure 3) were calculated using these reference parameters and for the two layers assumed to present variations in Young's modulus. Their shape is not intuitive, with both positive and negative parts depending on $x$, which partially offset one another in the calculation of indicators. The weighting functions of the two layers assume opposite shapes, yet with very different scales.

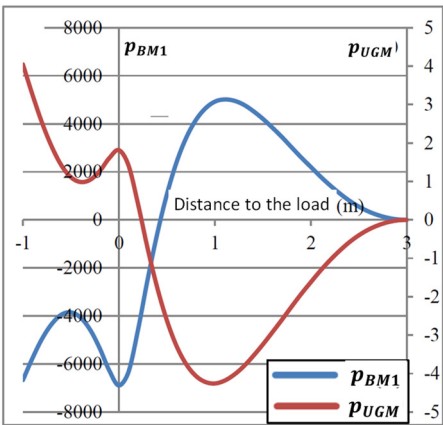

**Figure 3.** Weighting functions for the reference structure of Table 2; blue: function relative to the upper base layer (**left** scale); red: function relative to the subgrade layer (**right** scale).

Using these weighting functions, the two indicators $\Delta I_{\text{BM1}}$ and $\Delta I_{\text{UGM}}$ were calculated for the theoretical curviameter deflection bowls shown in Figure 2, as obtained for the various stiffness conditions (Figure 4). For the reference structure, the indicators equal 0. As expected, close to this situation, the indicators are roughly equal to the difference in Young's modulus between the considered structure and the reference structure. Such an observation is valid within the interval $s\left[\frac{2}{3}E_{ref}, \frac{3}{2}E_{ref}\right]$ for both the BM$_1$ ($E_{ref} = 9000$ MPa) and UGM ($E_{ref} = 50$ MPa) layers. We also confirmed that the subgrade layer indicator remains nearly constant in this interval with respect to large variations of the base layer stiffness (see orthogonality). This property has also been verified for the base layer indicator, but only for small variations in subgrade stiffness. This limitation is due to the high sensitivity of the deflection bowl relative to subgrade stiffness, as shown in Figure 2.

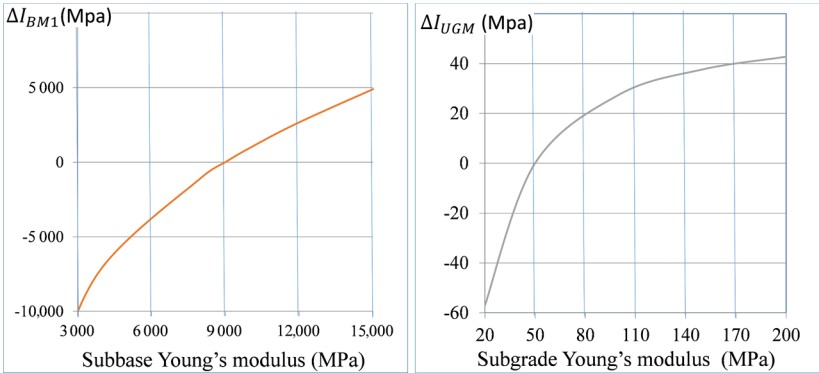

**Figure 4.** Evolution of indicators $(\Delta I_{\text{BM1}})$, $(\Delta I_{\text{UGM}})$ vs. the Young's modulus of the base and subgrade layers. The $y$ coordinate provides an estimate of the difference between the actual and reference stiffness moduli of the layers. These calculations have been performed for the theoretical deflection bowl of the curviameter. The left (resp. right) curve has been obtained for the reference modulus of the subgrade layer (resp. base layer).

The simulated curviameter deflection bowls have also been used to calculate the value of the various indicators in Table 1, for purposes of comparison with the indicators $\Delta I_{\text{BM1}}$ and $\Delta I_{\text{UGM}}$ (Figure 5). Here, each indicator has been normalized with respect to the values obtained for the minimum stiffness modulus values of BM$_1$ (3000 MPa) and UGM (20 MPa). For example

$$\text{BLI}_{\text{BM1 } norm} = \text{BLI}/\text{BLI}(E_{\text{BM1}} = 3000 \text{ MPa}, E_{\text{UGM}} = 50 \text{ MPa})$$

$$\text{BLI}_{\text{UGM } norm} = \text{BLI}/\text{BLI}(E_{\text{BM1}} = 9000 \text{ MPa}, E_{\text{UGM}} = 20 \text{ MPa})$$

$$\Delta I_{\text{BM1 } norm} = \Delta I_{\text{BM1}} / \Delta I_{\text{BM1}} (E_{\text{BM1}} = 3000 \text{ MPa}, E_{\text{UGM}} = 50 \text{ MPa})$$

$$\Delta I_{\text{UGM } norm} = \Delta I_{\text{UGM}} / \Delta I_{\text{UGM}} (E_{\text{BM1}} = 9000 \text{ MPa}, E_{\text{UGM}} = 20 \text{ MPa})$$

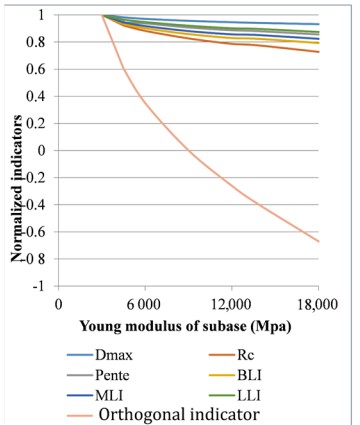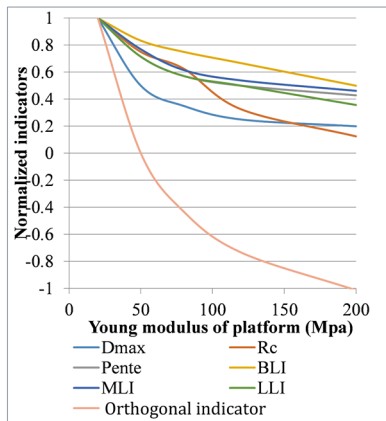

**Figure 5.** Theoretical comparison between the sensitivity of the conventional and orthogonal normalized indicators, relative to deflection measurements; **left**: sensitivity to the subbase stiffness modulus; **right**: sensitivity to the subgrade stiffness modulus.

Unsurprisingly, the optimized orthogonal indicator calculated for a layer is more sensitive to variations in the corresponding layer stiffness than conventional indicators. For the base layer, the conventional indicators only decrease by 20% of the relative layer stiffness increase, while the optimized indicator varies by 200% over this same interval [3000–18,000] MPa. The conventional indicators, which are more sensitive to variations in subgrade stiffness, display a sensitivity half that of the optimized indicator $\Delta I_{\text{UGM } norm}$ for a $E_{\text{UGM}}$ value lying between 20 and 200 MPa.

Furthermore, each of the conventional indicators is sensitive to variations in the stiffness modulus of both layers. It is thus difficult to define whether their variation is due to the foundation layer or the base layer. On the other hand, orthogonal indicators are especially sensitive to the stiffness of the layer for which they were designed and their overall consideration is able to directly designate the damaged layer.

### 3.1. Local Variations of E-Moduli (Theoretical Application Example)

To more fully illustrate the benefit of these optimized indicators, let us consider a 30 m long road section monitored with a curviameter. The reference structure is the same as above. As previously, the pavement includes theoretical stiffness variations in either the base (BM$_1$) or subgrade layer (UGM), but the variations here have the shape of a descending staircase as shown in Figure 6. Only small changes in the deflection bowls can be observed among the various stiffness conditions. The maximum deflection varies by just 10 μm in the first case and 50 μm in the second. Therefore, for conventional deflection indicators (Table 1), the pavement structure appears to be nearly homogeneous. Regardless of the conventional parameter chosen (e.g., Dmax, RoC, BLI, etc.), the variations are less than 2% for Young's modulus variations in the base layer and 5% for variations in the subgrade layer.

In contrast, Figure 7 shows the variations in the optimized indicators obtained for both stiffness profile cases. It can be observed that these indicators follow the evolution of the modulus for the layer concerned. Furthermore, the value of the orthogonal indicator corresponds to the difference in Young's modulus between the reference structure and the modified structure. This finding confirms that the new indicators can be directly used to estimate the physical parameters of the structure.

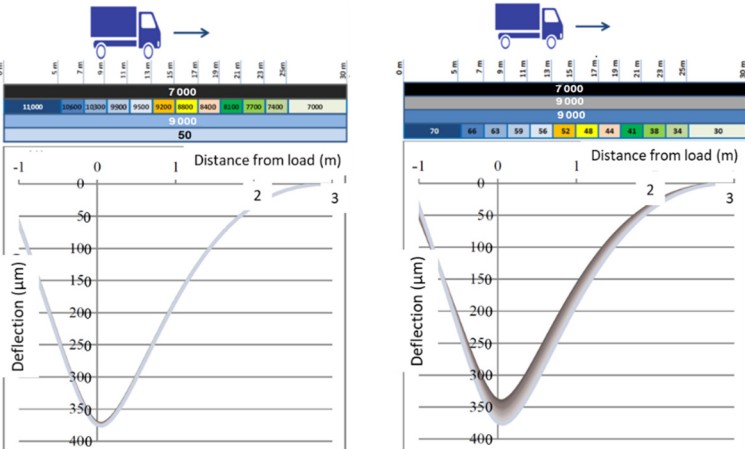

**Figure 6.** Theoretical curviameter bowls for the pavement structure presented above the graphs. The Young's modulus value is assumed to decrease every 2 m; **left**: variations in the base layer; **right**: variations in the subgrade layer. The deflection bowls have been computed for the geophone placed at the center of the areas with constant stiffness.

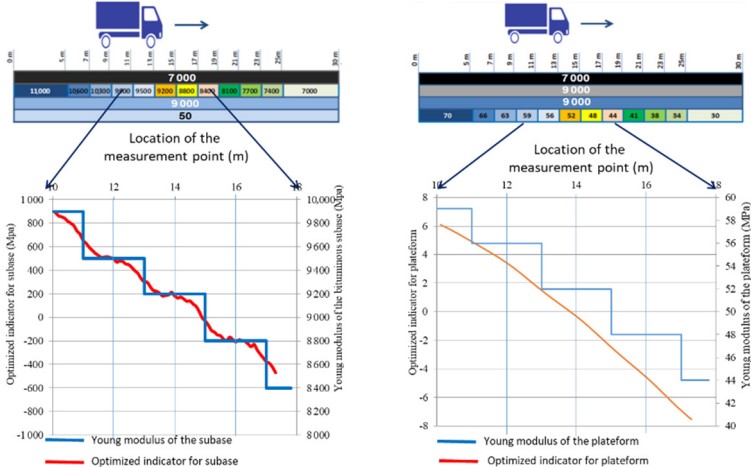

**Figure 7.** Simulation of the response of indicators $I_{BM1}$ (**left**) and $I_{UGM}$ (**right**) for the pavement structure presented above the graphs and for the deflection bowls in Figure 6.

### 3.2. Sensitivity of the Indicators to Measurement Errors

It should be noted that all calculations above have been carried out for the ideal case of measurements without error. As a matter of fact, the measurement noise is likely to degrade the performance of the indicators ($I_l$). This is true for both conventional indicators and the optimized ones proposed herein.

Conventional indicators are normally based on just one or two parameters of a signal, as depicted in Table 1 whereas optimized indicators make use of all available values. Thus, these latter indicators use seven or more deflection values when measured with a falling weight deflectometer, or the 100 points when measured with a curviameter. Using more measurement points in a signal will reduce indicator sensitivity to all individual noise. In this regard, the Appendix A shows that increasing, up to a certain extent, the number of measurements used to calculate the optimized indicators serves to reduce the effect of measurement uncertainties.

### 4. Possible Extensions to the Method

The method discussed above for constructing indicators can be extended in several ways in accordance with different objectives. Some sample possibilities are listed below;

they have been obtained by modifying the model used to calculate the deflection function $w_{\mathcal{M}}$. The models used must however avoid too many unknown parameters.

### 4.1. Model with Interface Shear Stiffness

Burmister's model and its semi-analytical solutions can be easily extended to the explicit presence of 'semi-sliding' interface condition(s), thus avoiding the introduction of additional thin layers. Indeed, elastic interface behavior between layers $l, l+1$ can be directly modeled using the interface shear stiffness $\kappa_{l,l+1}$ defined by the following equations:

$$\sigma_{xz} = \kappa_{l,l+1}\left(u^+ - u^-\right) \; and \; \sigma_{yz} = \kappa_{l,l+1}\left(v^+ - v^-\right) \tag{19}$$

where $u^+$, $u^-$, $v^+$, $v^-$ are the local horizontal displacements on both sides of the interface.

Denoting $w_{\mathfrak{M}}\left(x; \{E_k\}_u, \{\kappa_{j,j+1}\}_{int}\right)$ as the solution to Burmister's model with stiffness $\kappa_{j,j+1}$ for $j \in Index_{int}$, indicator(s) $I_{l,l+1}$ can be built in the same way as before, in particular using the (numerical) derivative(s) $\partial w_{\mathfrak{M}}/\partial \kappa_{j,j+1}$.

### 4.2. Visco-Dynamic Models for FWD or HWD Measurements

For FWD or HWD measurements, as presented in many references (see for example [11,21]), a significant improvement in the model $\mathfrak{M}$ can be obtained by considering the dynamic forces and possibly the viscoelastic behavior of the asphalt layers. In this case, the specific mass of the pavement materials can reasonably be approximated by reference values that depend on their category without the need to introduce new unknown numerical parameters. The modulus and phase angle of the complex modulus, which are necessary for the direct visco-dynamic model $w_{vd}$, can be approximated by the following function of the pulsation $\omega$

$$|E_l^*(\omega)| \approx a_l + b_l \ln \omega, \; \varphi \approx \frac{\pi}{2}\frac{d\, ln|E_l^*(\omega)|}{d\ln\omega} = \frac{\pi}{2}\frac{b_l}{a_l + b_l \ln \omega} \tag{20}$$

$w_{vd}$ can then be considered as a function of both time and coefficients $(a_l, b_l)$ for $l \in Index_u$ with $b_l = 0$ for non-bituminous materials.

For its part, the $w_{\mathfrak{M}}$ function can be built by retaining only the theoretical maximum deflection calculated for each geophone, i.e.,

$$w_{\mathfrak{M}}(x_i; a_l, b_l) = \underbrace{Max}_{over\ time\ t}\; w_{vd}(x_i, t; a_l, b_l) \tag{21}$$

in conjunction with the same operation performed on H/FWD deflection measurements.

### 4.3. Application to Structural Health Monitoring with Embedded Sensors

The onsite instrumentation of infrastructure is a common practice for their structural health monitoring. In recent years, this method has been developed for the local survey of pavement sections assumed to be representative of the behavior of a homogeneous route (similar pavement design, same traffic, same climatic conditions). Several types of embedded sensors are used, such as temperature sensors, strain gauges, geophones, and accelerometers delivering signals $\int_{meas}$ other than deflection. Infrastructure monitoring thus seeks to analyze the evolution of these signals over time or as a function of the cumulative traffic, in order to detect and assess the damage processes involved. However, while the standard data analysis reflects the overall evolution of pavement behavior, it does not in general make it possible to track the evolution of more specific parameters, such as the stiffness of a given layer.

The method described above could then be adapted to follow a more specific characteristic of the structure. In this case, the deflection data $w_{meas}(x; s_m)$ and functions $w_{\mathcal{M}}, w_{\mathcal{M}, E_k}$ should be replaced by the measured signals $\int_{meas}$ as well as by the modeled ones $\int_{\mathcal{M}}$ and their derivatives $\int_{\mathcal{M}, E_{jk}}$. The relationship between the time t of the recorded signals and

the load–sensor distance x should be deduced from either a specific measuring device also installed on the roadway, or the signal $\int_{meas}$ itself. The signals $\int_{meas}$ could be those induced by the traffic, by using statistical interpretation; or they could result from the passage of vehicles under calibrated conditions (wheel loads, position, speed, etc.).

## 5. Conclusions

This paper has proposed a methodology for calculating an optimized set of pavement (deflection) indicators, specially adapted to the structure under investigation. Weighting functions have been defined to calculate the indicators as linear forms of the deflection bowls. Each indicator is set up to be directly sensitive to one specific part of the structure without being sensitive to the others. These properties lead to a significant advantage over conventional indicators in evaluating a pavement structure or refining a diagnosis.

An initial theoretical example has been treated in this paper in order to illustrate the method, calculate the optimized indicators, and simulate their response. This example has confirmed two main benefits anticipated from the orthogonal indicators in comparison with the conventional ones. It could be verified that they are especially sensitive to the specific defect for which they have been designed. Moreover, their sensitivity is further improved by a factor above 2 compared with the conventional indicators. It can thus be considered that the use of such a deflection measurement interpretation technique is likely to facilitate pavement monitoring, by means of better identifying and locating defects along a given route.

The physical interpretation of indicator variations also constitutes a major advantage of the new indicators. Consequently, indicator variations along a road can be directly used to estimate physical parameter variations in the structure, such as Young's modulus. This feature will be very helpful for the road engineer to refine the diagnosis on the basis of a reference structure and then estimate the reinforcement works.

Possible extensions of the method have been suggested in this paper through the use of more advanced pavement mechanical models than Burmister's or other measurement systems. The method can be applied to the local survey over time or a number of loadings of a short, instrumented pavement section. Onsite applications are currently underway using our pavement testing facilities, including instrumented pavements with geophones or strain gauges. This campaign should confirm the real benefits of this approach for pavement monitoring.

The method is also being implemented in our software dedicated to pavements, one component of which pertains automatic back-calculation of measured deflection bowls at the scale of a road network. A major advantage expected over the more typical back-calculation techniques consists of better detecting correlations or correlation losses between the responses of adjacent pavement sections, built more or less under the same conditions and subjected to the same thermal and mechanical loadings.

**Author Contributions:** Conceptualization, J.-M.S., J.-M.P. and M.F.; methodology, J.-M.S., J.-M.P. and V.L.-B.; software, J.-M.P., V.L.-B. and M.F.; validation, J.-M.S., J.-M.P. and M.F.; formal analysis, J.-M.P.; investigation, J.-M.S., J.-M.P., V.L.-B. and M.F.; resources, J.-M.S., J.-M.P., V.L.-B. and M.F.; writing—original draft preparation, J.-M.P.; writing—review and editing, J.-M.S., J.-M.P., V.L.-B. and M.F.; funding acquisition, J.-M.S. and V.L.-B. All authors have read and agreed to the published version of the manuscript.

**Funding:** This research was funded by the French Ministry of Ecological and Solidarity Transition and the Pays de la Loire region.

**Institutional Review Board Statement:** Not applicable.

**Informed Consent Statement:** Not applicable.

**Data Availability Statement:** Numerical data are available on request from the corresponding author.

**Conflicts of Interest:** The authors declare no conflict of interest.

## Appendix A. Sensitivity of the Optimized Indicators to Deflection Measurement Uncertainties

Let us assume that the measurement data $w_{meas}(x)$ at the basin deflection scale are affected by measurement errors $err(x)$, namely: $w_{meas}(x) = \overline{w}_{meas}(x) + err(x)$, where $\overline{w}_{meas}(x)$ would be (the theoretical measurement) without error.

Next, let us consider the indicators, as defined in the main text, by scalar products $I^{(n)} = \sum_{i=1}^{n} p_i^{(n)} w_{meas}^{(n)}(x_i) = p^{(n)}.w_{meas}^{(n)}$ where $p^{(n)}$ and $w_{meas}^{(n)}$ are two vectors of dimension $n$. These may indeed be approximations of "integral" indicators $I = \int_{Deflection\ bowl} p(x) w_{meas}(x)\,dx$ through application of the trapezoidal rule, i.e.,

$$
\begin{aligned}
I \approx I^{(n)} &= \sum_{i=1}^{n} p_i^{(n)}\, w_{meas\ i}^{(n)} \\
&with\ p_i^{(n)} = c\, p(x_i) \\
(c = 1\ &for\ 2 \leq i \leq n-1,\ c = 0.5\ for\ i = 1\ and\ i = n) \\
&and\ w_{meas\ i}^{(n)} = w_{meas}(x_i)
\end{aligned}
\tag{A1}
$$

The intervals $\Delta x_i = x_{i+1} - x_i$ are assumed to be large enough to consider the measurement errors $err(x_i)$ independent of each other ($\Delta x_i \approx$ tens of centimeters in the case of FWD data).

For the sake of simplicity, let us also assume the errors $err(x_i)$ have the same standard deviation $\sigma_{err}$. It can then be shown that the standard deviation $\sigma(I^{(n)})$ on the value of indicator $I^{(n)}$ is a decreasing function of number $n$. Indeed, from the independence of errors $(x_i)'s$, we obtain

$$
\sigma^2\left(I^{(n)}\right) = \sum_{i=1}^{n} p_i^{(n)^2} \sigma_{err}{}^2 \Rightarrow \sigma\left(I^{(n)}\right) = \|p^{(n)}\|\, \sigma_{err}
\tag{A2}
$$

Hence, the smaller the norm of $\|p^{(n)}\|$, the smaller the value of $\sigma(I^{(n)})$. Let us now compare $\|p^{(n)}\|$ and $\|p^{(m)}\|$ for $n > m$ using the estimation below.

By construction, the indicators $I^{(n)}$ introduced in the main text have a norm given by

$$
\|p^{(n)}\| = 1/\|\overline{w}_{\mathfrak{M},\,E}\|^{(n)}
\tag{A3}
$$

where $E$ is the Young's modulus of the considered layer.

The calculation of $\frac{1}{\|\overline{w}_{\mathfrak{M},\,E}\|^{(n)}}$ must be performed on a case-by-case basis depending on the deflection device and number $n$ considered. As an example, Table A1 compares the values $\|p^{(2)}\|$ and $\|p^{(7)}\|$ obtained in the case of a FWD with 2 and 7 geophones. The pavement structure is assumed to be that described in Table 2, with the indicators being related to the BM and UGM layers. The ratio of uncertainty measurements expected between the 2 FWD configurations is approximately $\frac{1}{2}$ ($\approx p_{BM1}^{(FWD\ 7)}/p_{BM1}^{(FWD\ 2)}$) for the BM1 layer and $1/3$ ($= p_{UGM}^{(FWD\ 7)}/p_{UGM}^{(FWD\ 2)}$) for the UGM layer. More generally, in considering that $\|\overline{w}_{\mathfrak{M},\,E}\|^{(n)}$ has an order of magnitude of

$$
\|\overline{w}_{\mathfrak{M},\,E}\|^{(n)} \approx \left( \frac{n}{x_{max} - x_{min}} \int_{x_{min}}^{x_{max}} \overline{w}_{\mathfrak{M},\,E}{}^2(x)\,dx \right)^{1/2}
\tag{A4}
$$

the ratio $\|p^{(n)}\| / \|p^{(m)}\|$ and uncertainty of measurements are likely to vary as $\sqrt{m/n}$.

Table A1. Example of comparison between the norm of FWD indicators with 2 and 7 geophones.

| Weighting Function | Configuration with 2 Geophones Position of Geophones (cm) | | | | | | | Norm of Indicators |
|---|---|---|---|---|---|---|---|---|
| | G1 | G2 | | | | | | |
| | 0 | 30 | | | | | | |
| | Weighting coefficients | | | | | | | |
| $p_{BM1}^{(FWD\ 2)}$ | −550 | 566 | | | | | | 789 |
| | Weighting coefficients | | | | | | | |
| $p_{UGM}^{(FWD\ 2)}$ | 0.1568 | −0.2946 | | | | | | 0.33 |
| Weighting function | Configuration with 7 geophones Position of geophones (cm) | | | | | | | Norm of indicators |
| | G1 | G2 | G3 | G4 | G5 | G6 | G7 | |
| | 0 | 20 | 30 | 45 | 60 | 90 | 120 | |
| | Weighting coefficients | | | | | | | |
| $p_{BM1}^{(FWD\ 7)}$ | −249 | −88 | −17 | 49 | 95 | 145 | 159 | 357 |
| | Weighting coefficients | | | | | | | |
| $p_{UGM}^{(FWD\ 7)}$ | 0.0438 | 0.0010 | −0.0176 | −0.0345 | −0.0457 | −0.0567 | −0.0578 | 0.11 |

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
