# Peer review of "Orthogonal Set of Indicators for the Assessment of Flexible Pavement Stiffness from Deflection Monitoring: Theoretical Formalism and Numerical Study"

_remotesensing, doi:10.3390/rs14030500_

Round 1

Reviewer 1 Report

This manuscript presents an interesting study. 

Author Response

We thank you for your review and interest in our paper. We have made some changes on our submission in order to improve the research design. We hope to have clarified the paper’s main issues.

Reviewer 2 Report

The submitted manuscript reported an interesting work about the monitoring of pavements along routes is generally based on the use of indicators directly derived from the measurements. Prior to the final decision, there are some technical suggestions: 

1) In the literature review, the authors noted the in-field deflection measurement techniques and achievements of flexible pavement, but how are the related researches from the views of analytical and numerical modeling? During the review of your paper, I found two new articles from Google Scholar you can discuss these in the review section. Not limited to these two articles: ref-1) Man, J., Yan, K., Miao, Y., et al., 2021. 3D Spectral element model with a space-decoupling technique for the response of transversely isotropic pavements to moving vehicular loading. Road Materials and Pavement Design, pp.1-25. ref-2) Zhang, X., Otto, F. and Oeser, M., 2021. Modeling Pavement Surface Deflections under Accelerated Pavement Testing Using the PCA Method. Journal of Construction Engineering and Management, 147(12), p.04021169

2) In section 5, numerical applications of the method, theoretical examples, I am wondering how are the verifications of the model? The authors reported that the influence of modulus of base and subbase on the surface deflection or vertical displacements of pavement, makes sense, but how are you to make sure of the accuracy of the calculations? 

3) As mentioned by the authors that all calculations above are carried out for the ideal case of measurements without error. In fact, the measurement noise is likely to degrade the performance of the indicators. It is recommended to demonstrate these in detail in order to make the potential audiences more clear about that works. Meanwhile, the authors also noted that, in the Appendix that increasing, up to a certain point, the number of measurements used to calculate the optimized indicators makes it possible to reduce the effect of measurement uncertainties. Why? how to get this threshold for the applications? In general, again, it is an intersting research works, but I have some suggestions as mentioned above. Please respond and consider these first prior to the publication. 

Author Response

Dear reviewer,

We thank you for your review and interest in our paper.  We have modified our submission in order to consider your suggestions.

Thus, it has been proofread by a native English translator.

The “Conclusion” has been slightly developed. In particular, we mention that we are currently performing some tests on a short, instrumented pavement section in our accelerated testing facility (ALT) to apply and test the optimized deflection indicators method. These experiments should make it possible to better assess the benefit of the method on real data.

About point 1)

Thank you for the 2 references mentioned. The article “ref-1” is now cited in part 4 of our paper (Possible extensions to the method), since the model depicted therein could be used to simulate the dynamic effects generated by FWD or HWD in our analysis.

The approach depicted in “ref-2” based on the “response of real pavements + PCA analysis” could also be an interesting alternative to Burmister’s model. However, we would like to compare the results from this article with some of our own ALT experiments first so we can better assess the difference between both models. We did not add this reference to the current version of our paper.

About point 2)

Concerning the numerical part, the details and verification of the calculations are available in the thesis of Vinciane Le Boursicaud.

The Burmister’s model used in this thesis, as well as in our paper, is an analytical model, which allows the calculation of exact results. The characteristics chosen for the materials (soil, asphalt concretes) correspond to properties standardly used for pavement construction and can be measured by laboratory tests. The assumption of a rigid bedrock 6 m deep is also a common practice in road structure design. This assumption was verified by various tests (unpublished) in the 60’s by the use of sensors anchored at depth.  In general, it has been demonstrated many times that the calculation code and pavement datasets we use allow us to correctly reproduce the response of flexible pavements under moving loads with known speed and asphalt temperature conditions. The text has been adapted to better highlight the accuracy of these assumptions.

About point 3)

We are aware that measurement uncertainty is an important issue in the assessment of the efficiency of these new indicators. We have developed the corresponding paragraph “Sensitivity of the indicators to measurement errors” and the appendix by comparing the number of measurement points used for the calculation of the conventional indicators (ex: 2 points) to the number of measurement points considered in the computation of the new indicators. From the theoretical point of view, the expected accuracy improvement when using “n” data is proportional to the square root of “n”; Therefore, the higher the “n” the more accurate the results.

Forthcoming articles should confirm the accuracy improvement due to the new indicators on real cases using measurements recorded at our accelerated pavement facilities. 

With best regards.

J-M Simonin

Reviewer 3 Report

Journal: Remote Sensing

Title: Orthogonal set of indicators for the assessment of pavement stiffness from deflection monitoring

Comments:

This study proposed a methodology to define a set of orthogonal indicators adapted to the pavement structure being evaluated. The research topic is interesting and meaningful. Here are some suggestions provided to improve your manuscript.

  1. Abstract: Too simple and more crucial information should be supplemented.
  2. Sections 1-3 should be combined into one section (introduction), and the corresponding contents should be more concise.
  3. Table 1: the related references should be listed here.
  4. Section 4: The connections between different paragraphs should be strengthened. And the writing format is not comfortable for readers.
  5. In addition, the format of all tables and figures should be unite.
  6. Apart from the main findings, more recommendations for future works should be added in Conclusion part.
  7. The English writing and paper format should be significantly improved before publishing it.

Author Response

Dear reviewer

We thank you for your review and interest in our paper.  We have modified our submission in order to consider your suggestions.

  1. The abstract has been developed to better highlight the method’s main points.
  2. Sections 1-3 have been combined. Some adjustments have been made.
  3. References have been added to table 1.
  4. Section 4 (now section 2) has been adapted to ease reading. Transitions between paragraphs have also been reinforced.
  5. Tables and figures have been unified.
  6. The “Conclusion” has been slightly extended. In particular, we mention that we are currently performing some tests on a short, instrumented pavement section in our accelerated testing facility (ALT), to apply and test the optimized deflection indicators method. These experiments should make it possible to better assess the benefit of the method on real data.
  7. The paper has been proofread by a native English translator.

With best regards.

J-Michel Simonin

Reviewer 4 Report

  • The novelty and scientific contribution of this paper do not appear to be significant enough for further consideration for the journal.
  • The paper does not put the progress adequately it reports in the context of previous works, representative referencing and original discussion.
  • When presenting the state of the art, the authors should highlight limitations and explain the differences in relation to the presented proposal. They also should explain the novelty and impact of their proposal.
  • The authors should identify guidelines for further research.
  • The paper also lacks sufficient technical depth.
  • The overall scientific contribution of the article is not significant enough.

Author Response

Dear Reviewer,

We thank you for your review and interest in our paper.  We have modified our submission in order to consider your suggestions. We tried to highlight the novelty of the paper: the method proposes a new set of indicators, which are both orthogonal and allow a direct physical interpretation of their variation (i.e. Young’s modulus variation of a specific layer along a road section). The classical indicators do not have such properties. The conclusion has been slightly extended with perspectives for future research. In particular, we mention that we are currently performing some tests on a short, instrumented pavement section in our accelerated testing facility (ALT), to apply and test the optimized deflection indicators method. These experiments should make it possible to better assess the benefit of the method on real data.

With best regards

J-Michel Simonin

Round 2

Reviewer 3 Report

Most of my comments have been addressed carefully and it now can be published. Congratulations!

Author Response

Thanks for your suggestions.

My co-authors and I enjoyed them. They help us to improved our article.

We thank you for your congratulations. 

Best regards.

J-Michel Simonin

Reviewer 4 Report

My comments have been addressed and the paper has been improved. As a consequence, the article can be accepted.

Author Response

Thanks for your review and suggestions.

My co-authors and I enjoyed them. They help us to improve our article.

Thank you to accept the paper. 

Best regards.

J-Michel Simonin